



**Nitrate formation from heterogeneous uptake of dinitrogen pentoxide during a severe**
**winter haze in southern China**
Hui Yun[1], Weihao Wang[1], Tao Wang[1,*], Men Xia[1], Chuan Yu[1,2], Zhe Wang[1], Steven C.N.
Poon[1], Dingli Yue[3], Yan Zhou[3]
[1]Department of Civil and Environmental Engineering, The Hong Kong Polytechnic
University, Hong Kong, China
[2]Environment Research Institute, Shandong University, Jinan, China
[3]Guangdong Environmental Monitoring Center, State Environmental Protection Key
Laboratory of Regional Air Quality Monitoring, Guangzhou, China
*Correspondence to: Tao Wang (cetwang@polyu.edu.hk)
**Abstract:** Nitrate ($NO_3^-$) has become a major component of fine particulate matter ($PM_{2.5}$)
during hazy days in China. However, the role of the heterogeneous reactions of dinitrogen
pentoxide ($N_2O_5$) in nitrate formation is not well constrained. In January 2017, a severe haze
event occurred in the Pearl River Delta (PRD) of southern China during which high levels of
$PM_{2.5}$ (~400 μg m$^{-3}$) and $O_3$ (~160 ppbv) were observed at a semi-rural site (Heshan) in the
western PRD. Nitrate concentrations were up to 108 μg m$^{-3}$ (1 h time resolution), and the
contribution of nitrate to $PM_{2.5}$ reached nearly 40%. Concurrent increases in $NO_3^-$ and $ClNO_2$
(with a maximum value of 8.3 ppbv in 1 min time resolution) were observed in the first
several hours after sunset, indicating an intense $N_2O_5$ heterogeneous uptake on aerosols. The
formation potential of $NO_3^-$ via $N_2O_5$ heterogeneous reactions was estimated to be 39.7 to 77.3
μg m$^{-3}$ in the early hours (3 to 6 h) after sunset based on the measurement data, which could
completely explain the measured increase in the $NO_3^-$ concentration during the same time
period. Daytime production of nitric acid from the gas-phase reaction of OH + $NO_2$ was
calculated with a chemical box model built using the Master Chemical Mechanism (MCM
v3.3.1) and constrained by the measurement data. The integrated nocturnal nitrate formed via
$N_2O_5$ chemistry was comparable to or even higher than the nitric acid formed during the
daytime. This study confirms that $N_2O_5$ heterogeneous chemistry was a significant source of
aerosol nitrate during hazy days in southern China.
**Keywords:** $N_2O_5$, $ClNO_2$, nitrate, Pearl River Delta, southern China



## 1 Introduction

Severe haze in China has been a major concern of the regulatory and scientific communities in recent years. Nitrate was identified as an important component of $PM_{2.5}$ during hazy days in both summer and winter (e.g., Huang et al., 2014; Li et al., 2018; Pathak et al., 2009; Zhang et al., 2015). Moreover, the proportion of nitrate in $PM_{2.5}$ has increased steadily in the last decade due to the lagged control of $NO_x$ emissions compared to $SO_2$ (Fu et al., 2014; Geng et al., 2017; Qu et al., 2017; Reuter et al., 2014; Wang X et al., 2016). As a result, the concentrations of nitrate in $PM_{2.5}/PM_{1.0}$ were even higher than those of sulfate during some haze events (Ge et al., 2017; Li et al., 2017; Liu et al., 2015; Yang et al., 2017; Yue et al., 2015).

Nitrate is formed from $NO_x$ in both the daytime and nighttime. During the day, nitric acid ($HNO_3$) is produced through the gas-phase reaction between OH and $NO_2$ (R1), and this pathway is insignificant at night due to very low OH concentrations (e.g., Seinfeld and Pandis, 2016). The nitric acid can react with ammonia ($NH_3$) to form ammonium nitrate ($NH_4NO_3$), and an equilibrium can be reached for these three compounds between the gas phase and the particle phase (R2-3). In the nighttime, heterogeneous uptake of $N_2O_5$, which is formed from the reactions involving $O_3$, $NO_2$ and $NO_3$, becomes a source of nitrate and also produces gaseous $ClNO_2$ when chloride-containing aerosol is present (R4-7) (Finlayson-Pitts et al., 1989). This nitrate formation pathway is important only at night due to the fast photolysis of $NO_3$ during the day. Compared to the relatively well-understood formation of aerosol nitrate via the OH + $NO_2$ reaction, the contribution from $N_2O_5$ heterogeneous reactions has been poorly quantified due to the limited knowledge of key factors controlling the heterogeneous processes, such as the $N_2O_5$ uptake coefficient ($\gamma_{N2O5}$) and $ClNO_2$ yield ($\phi_{ClNO2}$) (Brown and Stutz, 2012; Chang et al., 2011).

(R1) $OH + NO_2 + M \rightarrow HNO_3 + M$

(R2) $HNO_3 \, (g) + NH_3 \, (g) \leftrightarrow NH_4NO_3 \, (s)$

(R3) $HNO_3 \, (g) + NH_3 \, (g) \leftrightarrow NH_4^+ \, (aq) + NO_3^- \, (aq)$

(R4) $NO_2 + O_3 \rightarrow NO_3$





(R5) $NO_2 + NO_3 + M \leftrightarrow N_2O_5 + M$
(R6) $NO_3 + VOCs \rightarrow$ products
(R7) $N_2O_5 + H_2O$ or $Cl^- (aq) \rightarrow (2-\phi) NO_3^- (aq) + \phi\ ClNO_2 (g)$
Model studies initially treated $\gamma_{N2O5}$ as a constant (0.03 to 0.1) (Dentener and Crutzen,
1993;Makar et al., 1998;Munger et al., 1998;Schaap et al., 2004;Wen et al., 2015;Xue et al.,
2014), and later utilized several parameterization schemes of $\gamma_{N2O5}$ and $\phi_{ClNO2}$ based on the
laboratory investigations of their dependence on aerosol compositions and aerosol water
content (Anttila et al., 2006;Bertram and Thornton, 2009;Davis et al., 2008;Evans and Jacob,
2005;Riemer et al., 2009;Riemer et al., 2003;Roberts et al., 2009). However, recent studies
found a significant discrepancy between the field-derived and parameterized $\gamma_{N2O5}$ and $\phi_{ClNO2}$
(McDuffie et al., 2018; Phillips et al., 2016; Tham et al., 2018; Wang X et al., 2017; Wang Z
et al., 2017; Zhou et al., 2018). These findings suggest that $N_2O_5$ uptake is more complicated
than previously thought and a better understanding of the uptake process is needed to improve
the prediction of nitrate and haze.
In addition to the modeling approach, field measurements of trace gases and aerosol
composition have been used to infer the contribution of $N_2O_5$ heterogeneous chemistry to
nitrate formation. Pathak et al. (2009) postulated the importance of $N_2O_5$ heterogeneous
reactions to the high aerosol nitrate observed in summertime in Beijing and Shanghai by
examining the variation of nitrate with the change in relative humidity (RH) and the
equilibrium between anions and cations in $PM_{2.5}$. Pathak et al. (2011) further investigated
nitrate formation using a coupled aqueous phase radical mechanism (CAPRAM) and a
gas-phase chemistry mechanism (RACM, without $ClNO_2$ chemistry). By constraining the
uptake coefficient of $N_2O_5$ in the range of 0.001 to 0.1, they reproduced the observed
enhancement of nitrate and suggested that $N_2O_5$ uptake in aerosols contributed to 50 to 100%
of the nighttime increase in nitrate. A similar method was used recently by Wen et al. (2018)
to simulate the summertime nitrate formation in the North China Plain (NCP), which
demonstrated the dominant contribution of $N_2O_5$ heterogeneous reactions to nighttime nitrate
formation. Based on the observed covariation of nitrate and RH, Wang et al. (2009)



speculated that $N_2O_5$ reactions dominated the nitrate formation on polluted days with high
$NO_2$ and $O_3$ in Shanghai. Neither $N_2O_5$ nor $ClNO_2$ was measured during these early
observation-based studies. A recent study (Wang H et al., 2017) inferred $\gamma_{N2O5}$ from the
measured $N_2O_5$ on four days in urban Beijing and estimated the lower limit of the formation
potential of aerosol nitrate assuming a unity $\phi_{ClNO2}$ because $ClNO_2$ was not measured. Their
result showed a comparable contribution to nitrate formation from the $N_2O_5$ heterogeneous
chemistry as from the daytime pathway of the $OH + NO_2$ reaction.
In the present study, $N_2O_5$, $ClNO_2$, the related chemical and meteorological parameters were
measured at a semi-rural site in the Pearl River Delta of southern China from Jan 2 to Jan 15,
2017. A severe haze event was observed during the field study with $PM_{2.5}$ reaching 400 μg m$^{-3}$
and $O_3$ up to 160 ppbv. $ClNO_2$, which is only known to be produced from $N_2O_5$ heterogeneous
uptake, reached up to 8.3 ppbv, which is the highest ever reported value and revealed
extremely active $N_2O_5$ chemistry during the episode. The concurrent measurements of $N_2O_5$,
$ClNO_2$ and aerosol nitrate provide better constraints for elucidating nighttime $NO_3/N_2O_5$
chemistry and aerosol formation. An overview of the measurement data was first presented.
The nighttime processes that led to the formation of nitrate (e.g., production of $NO_3$ and $N_2O_5$,
$N_2O_5$ uptake coefficient, $ClNO_2$ yield) were analyzed. The nighttime formation potential of
nitrate was estimated based on these data and compared to the measured increase in nitrate.
The daytime production of nitric acid via the $OH + NO_2$ reaction was calculated based on a
box model using the Master Chemical Mechanism (MCM v3.3.1) and compared to the
nighttime formation potential of nitrate.
**2 Methods**
**2.1 Site description**
The field observation was conducted at the Guangdong Atmospheric Supersite, a semi-rural
site located at Hua Guo Shan (HGS, 22.728°N, 112.929°E) in the southwest of the city of
Heshan from Jan 2 to Jan 15, 2017. As shown in Fig. 1, HGS is a hill with a height of 60 m
above sea level. All measurement instruments were located on the 4$^{th}$ floor of a four-story
building on the top of the hill. The observation site was located in the western PRD where the



economic activity and population density are much less compared to central PRD. There are
five main roads near the HGS site, including three national roads (G325, G94 and G15), and
two provincial roads (S272 and S270). The hill is covered by subtropical trees and surrounded
by similar hills within close range, and a few residents live at the foot of the hill with some
farmland in the area.
**2.2 Chemical ionization mass spectrometer**
$N_2O_5$ and $ClNO_2$ were simultaneously observed using a quadrupole chemical ionization mass
spectrometer (THS Instruments, Atlanta) which converted $N_2O_5$ and $ClNO_2$ to ion clusters of
$I(N_2O_5)^-$ and $I(ClNO_2)$ (Tham et al., 2016; Wang T et al., 2016). Iodide ions ($I^-$) were
produced by exposing a mixture of $CH_3I/N_2$ (0.3%v/v) to an alpha radioactive source, 210-Po
(NRD, P-2031-2000). $I(N_2O_5)^-$ and $I(ClNO_2)^-$ were generated from the reaction between
$I^-(H_2O)$ and $N_2O_5/ClNO_2$, and were detected at 235 and 208 m/z, respectively. The time
resolution for the measurement was approximately 10 s, and the derived data were later
averaged to a time resolution of 1 min for further analysis. Activated carbon packed in a filter
before the sampling inlet was used to determine the instrument background. The calibration of
$N_2O_5$ and $ClNO_2$ were carried out every afternoon at the site, and the standard gas of $N_2O_5$
was added into the ambient air every 3 h to check the changes of sensitivity. A more detailed
description of the operation method of the CIMS can refer to Wang T et al. (2016). The
detection limits of $N_2O_5$ and $ClNO_2$ were 7 pptv and 6 pptv (3 $\sigma$, 1 min-averaged data),
respectively. The uncertainty and the precision of the measurement was ± 25 % and 3%,
respectively.
The inlet of the CIMS instrument was set approximately 1.5 m above the roof with 6 m long
PFA-Teflon tubing as the sampling line. The total sampling flow was set as 11 standard liters
per minute (SLPM). Four SLPM were diverted into the CIMS, $O_3$ and $NO_x$ analyzer, and the
remaining part was evacuated directly from the system. The total residence time was less than
1 s in the sampling system. To reduce the influence of the tubing wall adhered with deposited
particles, we replaced the inlet tubing and fittings every day and washed them with an
ultrasonic bath.



## 2.3 Other measurements

Trace gases of CO, $SO_2$, $O_3$, $NO_x$, total reactive nitrogen ($NO_y$), nitrous acid (HONO), C2 to C10 non-methane hydrocarbons (NMHCs) and oxygenated hydrocarbons (OVOCs) were measured. CO was observed using a gas filter correlation analyzer (Thermo Model 48i). $SO_2$ was measured using a pulsed fluorescence analyzer (Thermo Model 43i). $O_3$ was determined using a UV photometric analyzer (Thermo, Model 49i). NO and $NO_2$ were detected with a special chemiluminescence instrument (Thermo, Model 42i). A photolytic converter only sensitive to $NO_2$ was equipped in this instrument (Xu et al., 2013). $NO_y$ was determined using a general chemiluminescence analyzer (Thermo, Model 42i-Y) which was equipped with a molybdenum oxide (MoO) catalytic converter. HONO was detected using a long path absorption photometer (QUMA, Model LOPAP-03) (Xu et al., 2015). NMHCs were determined using an online gas chromatograph (GC) coupled with a flame ionization detector (FID) and a mass spectrometer (MS). NMHCs were only measured from Jan 2 to Jan 8, 2017 due to the maintenance of the GCMS after Jan 8. OVOCs (e.g., formaldehyde, acetaldehyde, acetone, methyl ethyl ketone) were sampled with 2,4-dinitrophenylhydrazine cartridges every 3 h and were later analyzed with a high-performance liquid chromatography (HPLC) system (Cui et al., 2016).

Concentrations of $PM_{2.5}$ were detected with a multi-angle absorption photometer (MAAP, Thermo Model 5012). The ionic compositions of $PM_{2.5}$ were measured with an ion chromatography (GAC-IC) system equipped with a gas and aerosol collector at a time resolution of 30 min (Yue et al., 2015), and the data were also averaged every 1 h to meet the time resolution of other components of $PM_{2.5}$. Organic carbon (OC) and elemental carbon (EC) were measured with an online OC/EC analyzer (RT-4, SUNSET) with a time resolution of 1 h. A scanning mobility particle sizer (SMPS Model 3936L75, TSI) was used to determine the dry-state particle number size distribution, covering the size range from 16.5 to 1000 nm. The ambient (wet) particle number size distributions were estimated based on a size-resolved kappa-Köhler function considering the variation with the relative humidity (Hennig et al., 2005; Liu et al., 2014). Aerosol surface density was then derived using the ambient particle number size distribution (wet) and an assumption of spherical particles (Tham et al., 2016;





Wang Z et al., 2017).
Meteorological parameters were measured with a portable weather station (Model WXT520,
Vaisala, Finland), including temperature, relative humidity (RH), wind direction, wind speed,
and pressure. A pyranometer (CMP22, Kipp & Zonen B.V., Holland) was used to measure the
solar radiation and the data were then utilized to derive the photolysis frequency of $NO_2$ based
on the method of Trebs et al. (2009).
**2.4 Chemical box model**
To estimate the daytime formation of nitric acid via the reaction of OH + $NO_2$, an
observation-based chemical box model developed with the latest version of the Master
Chemical Mechanism v3.3.1 (Jenkin et al., 2003; Jenkin et al., 2015; Saunders et al., 2003)
and an updated chlorine (Cl) radical chemistry module (Xue et al., 2015) was utilized to
calculate the mixing ratio of OH radicals and the reaction rate of OH + $NO_2$. The integrated
production of nitric acid during the daytime was then calculated based on the simulation
results. The box model was constrained with the observation data every 10 min, including the
data of $N_2O_5$, $ClNO_2$, HONO, $O_3$, NO, $NO_2$, $SO_2$, CO, C2 to C10 NMHCs, OVOCs
(formaldehyde, acetaldehyde, acetone, and MEK), temperature, aerosol surface density and
$J(NO_2)$, which were first averaged or interpolated. Average concentrations of NMHC species
during the daytime (7:00 to 17:00) and nighttime (17:00 to 7:00 of the next day) are shown in
Table S1. A function considering the variation of the solar zenith angle (Saunders et al., 2003)
was used to calculate the photolysis frequencies of HONO, $O_3$ and other species in clear sky,
which were then corrected with the $J(NO_2)$ values in the real environment. The $J(ClNO_2)$ was
treated the same as in Tham et al. (2016). The lifetime of unconstrained species respect to the
physical loss was set as 8 h in a boundary layer of 1000 m depth (equivalent to $3.47 \times 10^{-5}$ s$^{-1}$)
in order to avoid their accumulation. The model was run from 0:00 of Jan 3 to 11:00 of Jan 8,
2017. To stabilize the intermediate species, the simulation for the first 24 h was repeated six
times.





**3 Results and discussion**
**3.1 Overview of the observation**
Figure 2 shows the time series of $N_2O_5$, $ClNO_2$, components of $PM_{2.5}$, related trace gases and
meteorological parameters from 18:40 of Jan 2 to 11:00 of Jan 15, 2017. The average
temperature and RH during the measurement period were $17 \pm 4\,°C$ and $86 \pm 14\%$,
respectively. A severe pollution episode occurred on Jan 3 to 7 due to stagnant meteorological
conditions (Fig. 3 (a)), and the concentrations of most pollutants decreased to very low levels
on Jan 9 and Jan 12 to 15, which corresponded to the change in weather conditions. The most
polluted days were Jan 5 and 6 with the highest $PM_{2.5}$ of 400 $\mu g\ m^{-3}$ and the highest $O_3$ of 160
ppbv. The $PM_{2.5}$ data from the PRD regional air quality monitoring network revealed that the
HGS site was within the most polluted area during this haze event (Fig. 3(b)). This pollution
event was characterized by concurrent high levels of $PM_{2.5}$ and $O_3$ and is in contrast to the
winter haze in north China, which experienced high $PM_{2.5}$ but low $O_3$ (e.g., Sun et al., 2016;
Wang H et al., 2018a). The mixing ratios of $N_2O_5$ and $ClNO_2$ were up to 3358 pptv and 8324
pptv (1 min time resolution), respectively, indicating active $N_2O_5$ heterogeneous chemistry.
Very high concentrations of aerosol nitrate (up to 108 $\mu g\ m^{-3}$, 1 h time resolution) were also
observed during the multi-day episode. Nitrate contributed to 24% of the total $PM_{2.5}$ mass
concentration on average, which was comparable to that of organic matters (OM = 1.7*OC,
28%) and much higher than that of sulfate (16%) and ammonium (11%). The contribution of
nitrate to the $PM_{2.5}$ increased with an increase in nitrate concentration, and reached nearly 40%
at its highest nitrate level, indicating that nitrate was a dominant component of the $PM_{2.5}$ on
the most polluted days. The concentration of $NO_3^-$ exhibited a concurrent increase with that of
$ClNO_2$ in the early nighttime on Jan 3 to 4, Jan 4 to 5, Jan 5 to 6 and Jan 9 to 10 (see Fig. 4),
suggesting that $N_2O_5$ heterogeneous reactions significantly contributed to the formation of
nitrate during the nighttime. The measured increases of the $NO_3^-$ concentration during these
four nights were 35.3, 50.9, 43.3 and 32.7 $\mu g\ m^{-3}$, respectively. A similar increase in $ClNO_2$
was observed on Jan 6 to 7, but the composition of the $PM_{2.5}$ was not available due to
instrument maintenance. The discussion in the remainder of this manuscript will focus on the
detailed analysis of these five nights to investigate the role of $N_2O_5$ heterogeneous chemistry





in nitrate formation.

### 3.2 $N_2O_5$ heterogeneous chemistry on the selected nights


### 3.2.1 Production of $NO_3$ and $N_2O_5$


The first step in the nighttime nitrate formation via $N_2O_5$ chemistry is the production of $NO_3$
and $N_2O_5$. To get insight into the key factors affecting the $NO_3/N_2O_5$ chemistry, the variation
of $N_2O_5$ and production rate of $NO_3$ were examined with some relevant gases and
meteorological parameters of the five nights. Fig. 5 shows the data of the night of Jan 4 to 5
as an example. Some common features were identified for all five nights. In general, low
wind speed ($< 2.0$ m s$^{-1}$) at night facilitated the accumulation of air pollutants, and high RH
was favorable for $N_2O_5$ heterogeneous uptake. In addition, high aerosol surface density
provided interfaces for $N_2O_5$ heterogeneous reactions.
In the first couple of hours after sunset (Fig 5, red rectangle), $N_2O_5$ exhibited a peak and
quickly dropped to hundreds of pptv, while nitrate and $ClNO_2$ concurrently increased, which
was indicative of the local production and loss of $N_2O_5$. NO was below the detection limit
during this period. The production rates of $NO_3$ ($P_{NO_3} = k_{NO_2+O_3}[NO_2][O_3]$) were the fastest
just after sunset and decreased gradually due to reduced $O_3$ levels. There was a period later in
the night (22:00 to 01:00) when fresh emissions of NO were observed, and the production of
$NO_3$ was suppressed due to the titration of $O_3$ by NO. In the later nighttime, NO was below
the detection limit (Fig. 5, blue rectangle). During this period, $NO_3$ and $N_2O_5$ were produced
at moderate rates, and the very low $N_2O_5$ concentrations (below the detection limit) suggested
a fast loss of $N_2O_5$ probably leading to the local production of $ClNO_2$ and nitrate, which was
not revealed in the observed variations of $ClNO_2$ and nitrate. The concentrations of $ClNO_2$
and nitrate during this period fluctuated due to the change in the air masses indicated by the
change in $SO_2$ concentrations and wind speeds.

### 3.2.2 $N_2O_5$ uptake coefficient and $ClNO_2$ yield


The $N_2O_5$ uptake coefficient and $ClNO_2$ yield, together with the reactivity of $NO_3$ with NO
and VOCs, determines the loss pathways of $NO_3$ and $N_2O_5$. To derive the uptake coefficient of
$N_2O_5$, a method suggested by McLaren et al. (2010) was applied by treating $NO_3$ and $N_2O_5$ as





a whole ($[NO_3] + [N_2O_5]$) without assuming the chemical system was in the steady state. This
approach considers that the change of $NO_3$ and $N_2O_5$ concentrations is mainly due to
$NO_3/N_2O_5$ chemistry, and thus it requires that the air mass have relatively stable chemical
conditions and not be subject to fresh NO emissions. It also requires that $ClNO_2$ is produced
from the $N_2O_5$ chemistry and has an increasing trend to derive the yield of $ClNO_2$. This
method is applicable for the early nighttime (red rectangle, section 3.2.1) for these five nights.
The variation rate of $[NO_3] + [N_2O_5]$ can be calculated by deducting the production rate of
$[NO_3] + [N_2O_5]$ with its loss rate as Eq. (1).
(1) $\frac{d([N_2O_5]+[NO_3])}{dt} = P_{NO_3} - L_{N_2O_5+NO_3}$
The loss of $[NO_3] + [N_2O_5]$ is through the $NO_3$ reaction with VOCs and $N_2O_5$ heterogeneous
reactions, which can both be expressed as pseudo first order losses as Eq. (2):
(2) $L_{N_2O_5+NO_3} = L_{NO_3} + L_{N_2O_5} = k_{NO_3}[NO_3] + k_{N_2O_5}[N_2O_5]$
where $k_{NO3}$ and $k_{N2O5}$ represent the total first order rate constants for $NO_3$ and $N_2O_5$,
respectively. The loss rate of $N_2O_5$ can then be obtained from Eq. (3):
(3) $L_{N_2O_5} = k_{N_2O_5}[N_2O_5] = k_{NO_2+O_3}[NO_2][O_3] - \frac{d[N_2O_5]}{dt} - \frac{d[NO_3]}{dt} - k_{NO_3}[NO_3]$
Because $NO_3$ was not measured, it was calculated by assuming an equilibrium of
$NO_2$-$NO_3$-$N_2O_5$ as shown in Eq. (4). High levels of NO would break this equilibrium. Thus,
the periods with detected NO were excluded. $d[NO_3]/dt$ and $d[N_2O_5]/dt$ were calculated as the
rate of change of $NO_3$ and $N_2O_5$ in a time resolution of 10 min. $k_{NO3}$ was derived with the
measured concentrations of NMHCs as Eq. (5) by interpolating the data of NMHCs to 10 min
time resolution. The $NO_3$ reactivity with VOCs ($k_{NO3}$) in the early nighttime ranged from
$0.632$ to $1.54 \times 10^{-3}\,s^{-1}$ (Table 1), which was higher than those derived at Mt. TMS in winter
2013 ($0.17$ to $1.1 \times 10^{-3}\,s^{-1}$) (Brown et al., 2016), but lower than those in the North China Plain
during the summertime ($2$-$57 \times 10^{-3}\,s^{-1}$) (Tham et al., 2016; Wang H et al., 2017, 2018b; Wang
Z et al., 2017). NMHCs were not measured on Jan 9 to 10, 2017. Therefore, we used the
average $k_{NO3}$ in the early nighttime on Jan 3 to 4 as a replacement because these two periods





had similar pollution levels for most pollutants. For the later nighttime (Fig. 5, blue rectangle),
low levels of $N_2O_5$ and moderate levels of $P_{NO3}$ also made Eq. (3) inapplicable even though
NO was not detected.
(4) $[NO_3] = \frac{[N_2O_5]}{[NO_2] \times K_{eq}}$
(5) $k_{NO_3} = \sum k_i [VOC_i]$
Finally, the uptake coefficient of $N_2O_5$ was derived using Eq. (6) for every 10 min and
averaged for the whole selected periods. In Eq. (6), $C_{N2O5}$ is the mean molecular speed of
$N_2O_5$, and $S_a$ is the aerosol surface density. The yield of $ClNO_2$ was derived from Eq. (7) by
dividing the integrated production of $ClNO_2$ ($[ClNO_2]_{max}$) to the integrated loss of $N_2O_5$ since
sunset.
(6) $k_{N_2O_5} = \frac{L_{N_2O_5}}{[N_2O_5]} = \frac{1}{4} C_{N_2O_5} S_a \gamma_{N_2O_5}$
(7) $\phi = \frac{[ClNO_2]_{max}}{\int L_{N_2O_5} dt}$
The relative importance of $NO_3$ reactions with VOCs and $N_2O_5$ heterogeneous reactions can
be examined by comparing the values of the loss coefficient of $NO_3$ reactions ($\frac{k_{NO_3}}{[NO_2] \times K_{eq}}$) and
$N_2O_5$ heterogeneous reactions ($k_{N2O5}$) (Tham et al., 2016). Based on the calculations, the
values of $\frac{k_{NO_3}}{[NO_2] \times K_{eq}}$ were $1.82 \times 10^{-5}$ to $6.07 \times 10^{-5}$ s$^{-1}$ (see Table 1), while that of $k_{N2O5}$ were
$3.78 \times 10^{-3}$ to $20.4 \times 10^{-3}$ s$^{-1}$, which was two orders of magnitude higher than that of $\frac{k_{NO_3}}{[NO_2] \times K_{eq}}$,
suggesting that $N_2O_5$ heterogeneous reactions were the dominant loss pathway of $NO_3$ and
$N_2O_5$.
The average $\gamma_{N2O5}$ and $\phi_{ClNO2}$ derived for the early night of the five cases are listed in Table 1.
The data show that the uptake coefficient ranged from 0.009 to 0.101, which was comparable
to the previous values derived at Mt. Tai Mo Shan (TMS) in Hong Kong (0.004 to 0.022)
(Brown et al., 2016) and in the North China Plain (0.006 to 0.102) (Tham et al., 2018; Tham
et al., 2016; Wang H et al., 2017, 2018b; Wang X et al., 2017; Wang Z et al., 2017; Zhou et al.,
2018). The yield in this study varied from 0.20 to 0.36, which was similar to most studies in





China (Tham et al., 2018; Tham et al., 2016; Wang Z et al., 2017; Yun et al., 2018; Zhou et al.,

306    2018).

**3.3 Nitrate formation potential (pNO$_3^-$) through N$_2$O$_5$ chemistry**

**3.3.1 Nighttime p(NO$_3^-$)**

The formation potential of NO$_3^-$ through N$_2$O$_5$ chemistry is the total amount of NO$_3^-$
accumulated from N$_2$O$_5$ heterogeneous loss. It can be calculated by deducting the integrated
loss of N$_2$O$_5$ with the integrated production of ClNO$_2$ as Eq. (8).
(Eq.8)  $p(NO_3^-) = (2-\phi) \int L_{N_2O_5} \, dt = 2 \int L_{N_2O_5} \, dt - [ClNO_2]_{max}$
In the early nighttime, the average loss rate of N$_2$O$_5$ ($L_{N2O5}$) ranged from 1.9 to 3.9 ppbv h$^{-1}$
(Table 1), which was close to the average $P_{NO3}$ due to the dominance of the N$_2$O$_5$
heterogeneous reactions in NO$_3$ and N$_2$O$_5$ loss. Based on the derived N$_2$O$_5$ loss rate and the
maximum ClNO$_2$ concentration, the formation potential of NO$_3^-$ was derived and ranged from
39.7 to 77.3 μg m$^{-3}$ as shown in Fig. 6. The measured increase of the NO$_3^-$ concentration in the
early nighttime can be completely explained by the integrated production of NO$_3^-$ via the
N$_2$O$_5$ heterogeneous reactions during the same period.
In the later nighttime, the method described in section 3.2.2 was not valid for calculating the
N$_2$O$_5$ heterogeneous loss rate as mentioned above. We attempted to estimate the formation
potential of nitrate by assuming that the N$_2$O$_5$ heterogeneous reactions continued to dominate
the loss of NO$_3$ + N$_2$O$_5$ in the later nighttime. The $k_{NO3}$ in the later nighttime were comparable
to those in the early nighttime, and the high RH close to 100% in the later nighttime was
favorable for the N$_2$O$_5$ heterogeneous reactions. We assumed that all NO$_3$ was quickly
consumed by the N$_2$O$_5$ heterogeneous reactions, which means that the loss rate of N$_2$O$_5$
approximated to the production rate of NO$_3$ ($L_{N2O5} \approx P_{NO3}$). As listed in Table 2, the N$_2$O$_5$ loss
rates ranged from 0.82 to 1.26 ppbv h$^{-1}$, which were significantly lower than those derived in
the early nighttime. The derived N$_2$O$_5$ loss rate here and the yield of ClNO$_2$ in the early
nighttime were used to estimate the formation potential of NO$_3^-$ in the later nighttime. As
shown in Fig.6, the nitrate produced during these later periods ranged from 7.3 to 37.7 μg m$^{-3}$,





which was significantly lower than those in the early nighttime, indicating that the nighttime
nitrate from $N_2O_5$ chemistry was mainly produced in the early nighttime.

**3.3.2 Comparison with daytime production of $HNO_3$**

During the daytime, the formation of $NO_3^-$ is mainly from the gas-particle partitioning of the
gas phase $HNO_3$ formed through the $OH + NO_2$ reaction. Hence, the daytime formation
potential of $HNO_3$ ($p(HNO_3)$) can be treated as the upper limit for the locally-produced
daytime aerosol nitrate. To calculate the daytime $p(HNO_3)$, a box model based on MCM
v3.3.1 was used to derive the mixing ratio of OH and the rates of $OH + NO_2$ as described in
section 2.4. This model was previously used in our study at Wangdu in North China (Tham et
al., 2016). The calculated mixing ratios of OH at Wangdu with this model compared well with
those observed by the laser-induced fluorescence (LIF) technique (Tan et al., 2017). In the
present study, the average daytime OH (7:00 to 17:00) mixing ratios were 1.71 to $3.82 \times 10^6$
$cm^{-3}$ during Jan 3 to 7 as listed in Table 3 with the maximum values reaching 3.24 to $6.71 \times 10^6$
$cm^{-3}$. The detailed results for OH can be found in Fig. S1.
The average production rates of $HNO_3$ through the $OH + NO_2$ reaction were 1.40 to 5.21 ppbv
$h^{-1}$ from Jan 3 to Jan 7, and the integrated formation potential of $HNO_3$ during the daytime
was 35.7 to 131.8 µg $m^{-3}$, which was comparable to the nighttime $p(NO_3^-)$ ranging from 77.4
to 102.9 µg $m^{-3}$ (Fig. 7). Nighttime production of nitrate via the heterogeneous uptake of $N_2O_5$
accounted for 43.8 to 57.7% of the total nitrate ($NO_3^- + HNO_3$) produced in a 24 h period at
the site. These results underscored the important role of $N_2O_5$ heterogeneous chemistry in
nitrate formation in this severe winter haze in southern China.

**4 Concluding remarks**

With the use of concurrent measurements of nitrate, $ClNO_2$ and related pollutants, this study
demonstrates the important contribution of $N_2O_5$ heterogeneous uptake in nitrate formation.
Current chemical transport models have difficulties in simulating this nitrate production
pathway. Therefore, more research efforts are needed to improve the representations of $\gamma_{N2O5}$
and $\phi_{ClNO2}$ for better prediction of nitrate in the models. The observation-based approach
presented here can be applied to investigate nitrate formation in other areas of China.




**5 Data availability**
The data used in this study are available from the corresponding author upon request
(cetwang@polyu.edu.hk & dingliyue@163.com).
**Acknowledgment**
The authors thank Dr. Li Qinyi and Dr. Fu Xiao for helpful discussions, and Miss Yaru Wang
and Yiheng Liang for their help in analyzing the OVOC and aerosol composition. This study
was supported by the Hong Kong Research Grants Council (HK-RGC; C5022-14G and
PolyU 153026/14P) and National Natural Science Foundation (NNSF) of China (91544213).
Z. Wang acknowledges the support of the NNSF of China (41505103) and HK-RGC

(25221215).

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





Table 1. Average values of $N_2O_5$ concentrations, $N_2O_5$ uptake coefficients, $ClNO_2$ yields and
other related parameters and maximum values of $ClNO_2$ concentrations in the early nighttime

for five selected nights.

| Date | $N_2O_5$ | Max-$ClNO_2$ | $NO_2$ | $O_3$ | RH | Sa | $P_{NO3}$ | $k_{NO3}$ | $L_{N2O5}$ | $k_{NO3}/(Keq[NO_2])$ | $k_{N2O5}$ | $\gamma_{N2O5}$ | $\phi_{ClNO2}$ |
|---|---|---|---|---|---|---|---|---|---|---|---|---|---|
| | pptv | pptv | ppbv | ppbv | % | $\mu m^2\,cm^{-3}$ | ppbv $h^{-1}$ | $10^{-3}\,s^{-1}$ | ppbv $h^{-1}$ | $10^{-5}\,s^{-1}$ | $10^{-3}\,s^{-1}$ | | |
| Jan.3 17:40-20:50 | 102 | 3145 | 22 | 68 | 68 | 3644 | 4.0 | 0.632 | 3.9 | 3.26 | 20.4 | 0.101 | 0.36 |
| Jan 4 17:00-22:00 | 700 | 4608 | 24 | 61 | 82 | 6452 | 3.3 | 1.54 | 3.2 | 6.07 | 4.16 | 0.009 | 0.32 |
| Jan 5 17:00-22:00 | 338 | 4828 | 18 | 73 | 81 | 8399 | 3.4 | 0.790 | 3.3 | 4.06 | 9.00 | 0.015 | 0.29 |
| Jan 6 17:00-22:40 | 326 | 2908 | 13 | 82 | 77 | 5092 | 2.8 | 0.677 | 2.6 | 4.95 | 3.78 | 0.013 | 0.20 |
| Jan 9 19:00-00:20 | 121 | 2553 | 19 | 41 | 85 | 5173 | 1.9 | 0.632 | 1.9 | 1.82 | 4.28 | 0.015 | 0.28 |


Table 2. Average values of $N_2O_5$ loss rate and related parameters for selected periods in the

later nighttime.

| Date | | $NO_2$ | $O_3$ | $P_{NO3}$ | $k_{NO3}$ | $L_{N2O5}$ |
|---|---|---|---|---|---|---|
| | | ppbv | ppbv | ppbv $h^{-1}$ | $10^{-3}\,s^{-1}$ | ppbv $h^{-1}$ |
| Jan 3-4 | 21:00-05:00 | 20.8 | 20.7 | 1.00 | 0.684 | 1.00 |
| Jan 5 | 01:30-06:50 | 22.4 | 19.5 | 0.96 | 1.45 | 0.96 |
| Jan 5-6 | 23:40-01:10 | 21.1 | 25.5 | 1.26 | 1.13 | 1.26 |
| Jan 6-7 | 23:00-06:00 | 22.1 | 14.4 | 0.82 | 0.709 | 0.82 |
| Jan 10 | 01:50-03:30 | 24.8 | 15.6 | 0.90 | / | 0.90 |


Table 3. Average OH mixing ratio and rate of OH + $NO_2$ during the daytime (7:00 to 17:00 LT)

from Jan 3 to Jan 7, 2017.

| Date | OH | $NO_2$ | OH + $NO_2$ |
|---|---|---|---|
| | $(cm^{-3})$ | (ppbv) | (ppbv $h^{-1}$) |
| Jan 3 | $2.18\times10^6$ | 36.2 | 3.49 |
| Jan 4 | $2.47\times10^6$ | 23.6 | 2.60 |
| Jan 5 | $2.62\times10^6$ | 30.8 | 3.09 |
| Jan 6 | $3.82\times10^6$ | 31.5 | 5.21 |
| Jan 7 | $1.71\times10^6$ | 18.4 | 1.40 |








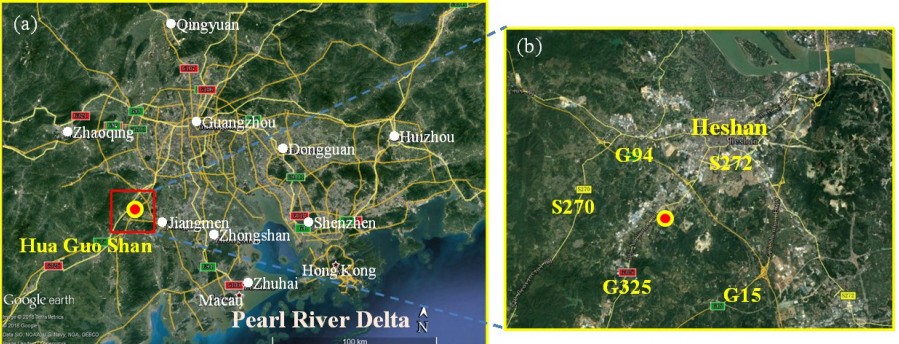


Figure 1. (a) Google map images of the Pearl River Delta in the Guangdong Province and
measurement site (Hua Guo Shan). (b) The topography and major roads (shown by number)
adjacent to the measurement site.

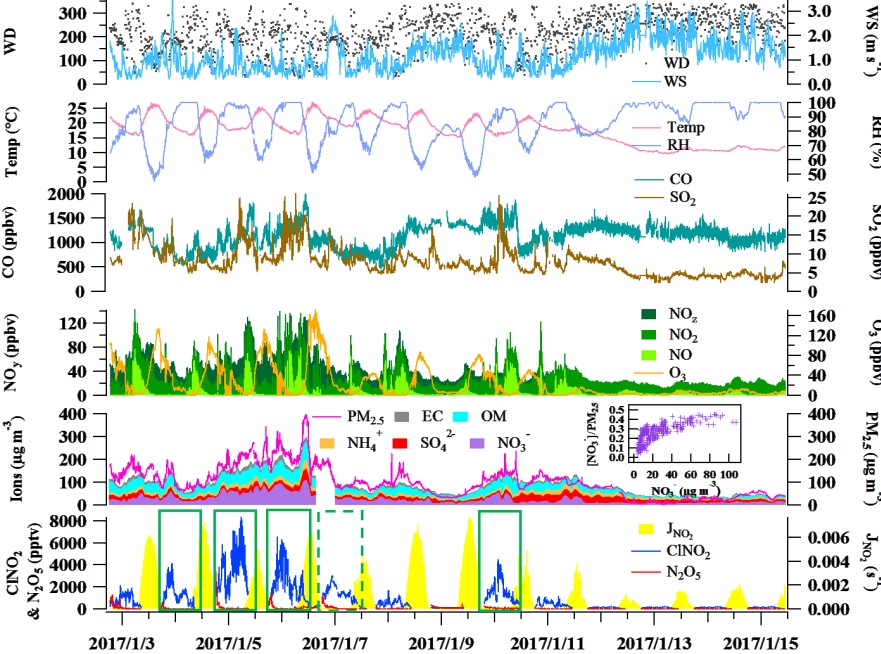


Figure 2. Time series of $N_2O_5$, $ClNO_2$, components of $PM_{2.5}$, related trace gases and
meteorological parameters from 18:40 of Jan 2 to 11:00 of Jan 15, 2017. The inserted figure
shows the variation of the ratio of nitrate to $PM_{2.5}$ with increasing nitrate concentration. The
green rectangles in the figure indicate the five days used for detailed analysis.





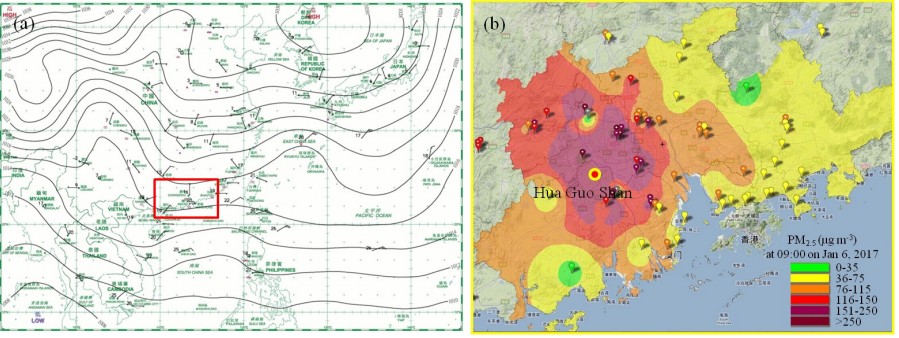


Figure 3. (a) Surface weather chart at 08:00 LT on Jan 6, 2017 downloaded from the website

of the Hong Kong Observatory indicating stagnant conditions. (b) The distribution of $PM_{2.5}$

concentrations in the PRD region at 09:00 LT on Jan 6, 2017. This figure was captured from

the website. http://113.108.142.147:20031/GDPublish/publish.aspx.

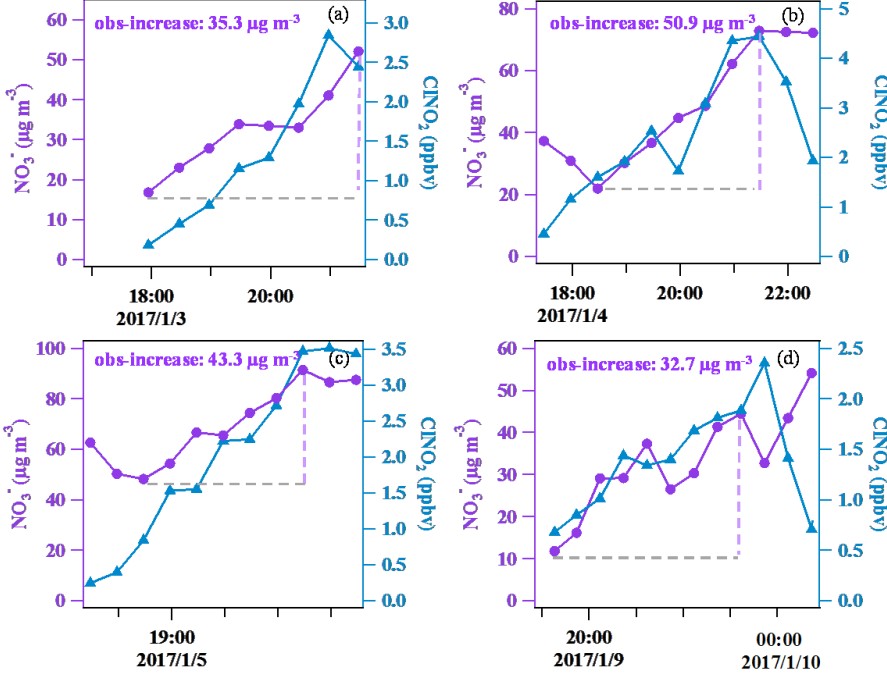


Figure 4. The covariance of aerosol nitrate and $ClNO_2$ in the early nighttime (in 30 min time

resolution) for four nights.



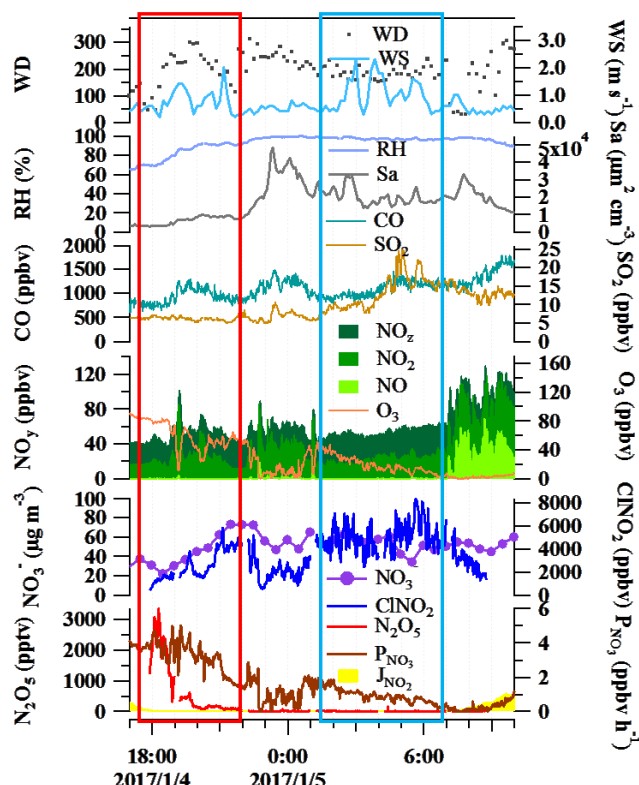


Figure 5. Variation of $N_2O_5$, $ClNO_2$, $NO_3^-$, trace gases and meteorological conditions during

the nighttime of Jan 4 to 5, 2017 as an example for the five selected nights.

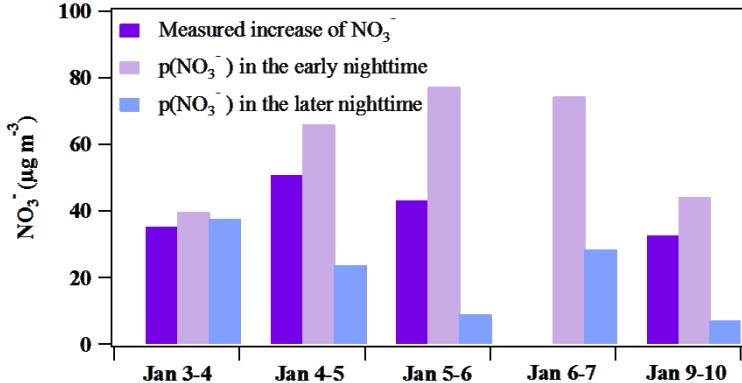


Figure 6. Comparison between the measured $NO_3^-$ increase and the $NO_3^-$ formation potential
in the early nighttime (periods in Table 1) and in the later nighttime (periods in Table 2).



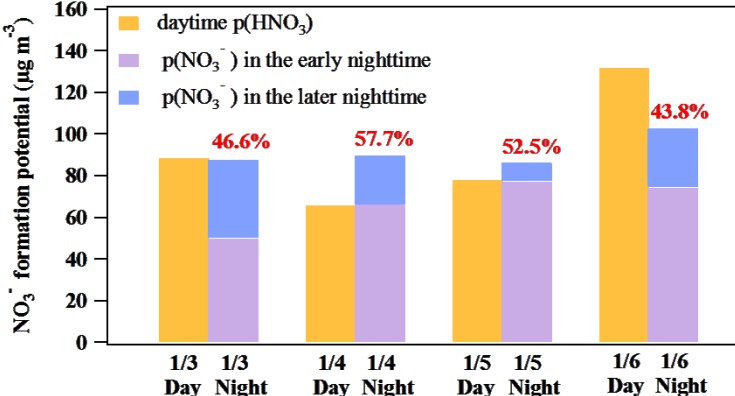


Figure 7. Comparison between the daytime (7:00 to 17:00 LT, assuming all gas phase $HNO_3$
partitioned into particle phase) and nighttime (17:00 to 7:00 LT of the next day) $NO_3^-$
formation potential.