# Peer review of "Nitrate formation from heterogeneous uptake of dinitrogen pentoxide during a severe"

_Atmospheric Chemistry and Physics, 2018_

## Referee Comment (RC1) · Anonymous Referee #1 · 23 Sep 2018

The manuscript of Yun et al., reported half month measurement of N2O5, ClNO2 and other relative parameters during heavy haze episodes in Pearl River Delta (PRD) of southern China. The N2O5 uptake coefficient and ClNO2 yield were determined from the observations. The study showed the observation evidence of the enhancement of particulate nitrate in the first several hours can be fully explained by the N2O5 heterogeneous hydrolysis and even comparable with the nitric acid formed by OH+NO2 during daytime. Overall, the paper contributes to the knowledge of N2O5 heterogeneous chemistry and highlight the heterogeneous reactions in the formation of particulate nitrate in southern China. The following comments should be addressed before publishing on ACP.

[Figure]

Major comments: The steady state assumption to derive the N2O5 uptake coefficient needs to be verified by model simulations under the observed conditions (with input from NO, NO2, O3, VOCs). It is useful to try other method (e.g. Brown et al., 2006) to derive N2O5 uptake coefficient. Brown, S. S., Ryerson, T. B., Wollny, A. G., Brock, C. A., Peltier, R., Sullivan, A. P., Weber, R. J., Dube, W. P., Trainer, M., Meagher, J. F., Fehsenfeld, F. C., and Ravishankara, A. R.: Variability in nocturnal nitrogen oxide processing and its role in regional air quality, Science, 311, 67-70, DOI 10.1126/science.1120120, 2006.

The uncertainty of the measured N2O5, NMHC and Sa and the overall uncertainty propagated to N2O5 uptake coefficient and ClNO2 yield should be carefully evaluated and discussed. As the hygroscopic growth factor is hard to quantify for RH over 90%, the derived N2O5 uptake coefficient for those conditions may subject with larger uncertainties compared with other RH cases. This is an interesting point to be discussed.

The relationship between N2O5 uptake coefficient, ClNO2 yield and the chemical properties of particles or the meteorological data (such as RH) should be investigated, especially in the part of text around Line 572 (table 1), the reason of the high gamma value in the Jan.3 17:40-20:50 should be addressed as which was much higher than other derived value.

Minor comments: The description of the experimental setup of the key relevant parameters needs to be strengthened, e.g. the limit of detection, the measurement uncertainties and measurement principle should be described.

Line 161. The reference of Yue et al., 2015 may not appropriate and suggests replacing by Dong et al., 2012 Dong, H. B., Zeng, L. M., Hu, M., Wu, Y. S., Zhang, Y. H., Slanina, J., Zheng, M., Wang, Z. F., and Jansen, R.: Technical Note: The application of an improved gas and aerosol collector for ambient air pollutants in China, Atmos Chem Phys, 12, 10519-10533, 10.5194/acp-12-10519-2012, 2012.

Line 586 (figure 2) please check the data of wind speed, as the WS keep below 3 m

s-1 during the whole half month. And the plot style of NOy made the concentration of NO2 hard to follow. The left y-axis of fourth panel should change to PM2.5 or other more appropriate name.

The legend of the early night and later night in figure 6 and 7 should be explained. By the way, how about the NO3- formation potential intercomparison in the day and night of Jan 9 to 10.

---

## Referee Comment (RC2) · Anonymous Referee #2 · 25 Sep 2018

Yun et al. present a suite of measurements related N2O5 formation and subsequent uptake to aerosol that take place in a semi-rural area of China. They show through interpretations of their measurements and some master chemical modeling that nocturnal NOx chemistry can likely account nearly 50% of aerosol nitrate mass loadings during these heavy pollution events.

This paper is written and presented well for the most part. The measurement methods portion is lacking even considering that an associated reference may describe additional details. Assuming my comments are appropriately addressed and some changes are made that would help to clarify the methods and the paper in general, I

would recommend publication.

Comments:

Line 97: recommend changing "highest ever reported value" to something that will age better like "largest reported value to date".

Line 100: recommend changing "aerosol formation" to "aerosol nitrate formation"

Line 128: the SI would be a great place to see the results of these instrument backgrounds and the extent to which they worked.

Line 129: "standard gas of N2O5" sounds like you can purchase a standard cylinder of N2O5 (which you can't). Even if Wang et al. 2016 outlines these calibrations in more detail, a brief explanation is needed at the minimum. The description of these calibrations needs to be expanded and include ClNO2 calibrations as well.

Line 132-133: How were detection limits calculated? What signal-to-noise was used, etc.? I think the authors only mean the uncertainty is +/- 25% not the precision.

Line 169: Here and throughout the paper it's probably best to change "aerosol surface density" to "aerosol surface area density" for clarity's sake.

Line 182: consider changing "calculate" to "estimate"

Line 214: change "matters" to "matter"

Line 265/284: k' is often used for a pseudo first order rate constant. Consider using that to help differentiate from other rate constants.

Line 307/309: make pNO3- and p(NO3-) consistent. Use one or the other. p(NO3-) is used in the rest of the paper.

Line 313/320 and Figure 6: Do the authors have a firm definition of what is considered "early nighttime" vs "late nighttime"? What times correspond to each period? Are these the same as provided in Table 1?

Line 343-345: please change "cm-3" units to commonly used "molec cm-3".

Line 359: certainly this approach is relevant to areas outside of China as well? Suggest removing "China".

Table 1: Addition average aerosol nitrate loadings and PM2.5 loadings for these periods would very useful. Consider adding all aerosol data (sulfate, ammonium, OM, etc.).

Figure 5: Why not include the other aerosol data in this figure? NO3- does not track with Sa, so what is driving up Sa? The other data should explain this.

Have the authors considered boundary layer effects in any of their analyses? With a shallow nighttime inversion layer and little mixing many of these species could be further concentrated. Are there any measurements taken during the study that would give boundary layer information (sondes, etc.)?
* * *

---

## Referee Comment (RC3) · Anonymous Referee #3 · 30 Sep 2018

The manuscript "Nitrate formation from heterogeneous uptake of dinitrogen pentoxide during a severe winter haze in southern China" by Yun and Co-authors uses observations of aerosol and gases and model results to study the contribution of heterogeneous chemistry via N2O5 to nitrate formation in PM2.5 during severe winter haze episodes. The measurements were carried out in the rural site of Hesnan, located near the Perl River Delta in Southern China. In addition to a comprehensive suite of measurements, Yun and co-authors present the results of a chemical box model to estimate daytime HNO3 mixing ratios. The box model was constrained by observatiosn and its results were used to assess the importance of nighttime N2O5 over daytime HNO3 as source of secondary aerosol nitrate.

The paper is well written, well structured and conveys results of interest for the scientific community. However, the method section (as pointed out by Refereees #1 and #2 as well) has to be improved and that some more discussion on 1)uncertainties, 2)sensitivity test of the model and 3)boundary layer dynamic needs to be added before publication.

1) The description of the measurements should include the detection limits and the uncertainties, in particular for the species that were used to constrain the chemical box model. 2) There should be a discussion in the main text or in the SI about the sensitivity of the box model to the uncertainties of the measurements (this, for example should be communicated with uncertainty bars in figure 7). 3) In paragraph 2.2 a discussion about interference for species with the same nominal mass as I(N2O5)- and I(ClNO2)- should be added. How much contribution from other species would Yun and Co-authors expect? If it was not negligible how would change the results from the box model/comparison? 4) Relative humidity (RH) is known to affect measurements carried out with I-CIMS. Was the inlet used in this study humidified? Was the RH controlled/monitored during zero measurements? How could the zero affect the box model results (e.g., over/under estimation of N2O5/ClNO2)? What are the biases that the 6 m sampling line could generate in their results? 5) The boundary layer plays a significant role in the time evolution of the concentrations of nitrate in the particle. Yun and Co-authors make little to no mention of its role. For example, one might expect that particulate nitrate would increase also in the early morning hours due to the contribution of the residual layer during the mixing. This doesn't seem to happen in the observations presented in this work. May the Authors discuss why that would be the case? 6) An increase in particulate nitrate concentrations (as well as PM2.5) could also be due to a dilution effect (same magnitude of aerosol sources but reduced volume in which the aerosols are mixed). I recommend adding a few sentences explaining how the mixing of the residual layer in the morning hours could affect the results presented here.

---

## Author Comment (AC1) · 12 Nov 2018

**The reviewers' comments are italicized followed by our responses and changes in manuscript shown in blue and red, respectively. And the corrections are also marked as red color in the revised manuscript.**

*The manuscript of Yun et al., reported half month measurement of $N_2O_5$, $ClNO_2$ and other relative parameters during heavy haze episodes in Pearl River Delta (PRD) of southern China. The $N_2O_5$ uptake coefficient and $ClNO_2$ yield were determined from the observations. The study showed the observation evidence of the enhancement of particulate nitrate in the first several hours can be fully explained by the $N_2O_5$ heterogeneous hydrolysis and even comparable with the nitric acid formed by $OH+NO_2$ during daytime. Overall, the paper contributes to the knowledge of $N_2O_5$ heterogeneous chemistry and highlight the heterogeneous reactions in the formation of particulate nitrate in southern China. The following comments should be addressed before publishing on ACP.*

Response: We appreciate the reviewer for the positive comments which are addressed in detail below.

*Major comments:*

*The steady state assumption to derive the $N_2O_5$ uptake coefficient needs to be verified by model simulations under the observed conditions (with input from NO, $NO_2$, $O_3$, VOCs). It is useful to try other method (e.g. Brown et al., 2006) to derive $N_2O_5$ uptake coefficient.*

*Brown, S. S., Ryerson, T. B., Wollny, A. G., Brock, C. A., Peltier, R., Sullivan, A. P., Weber, R. J., Dube, W. P., Trainer, M., Meagher, J. F., Fehsenfeld, F. C., and Ravishankara, A. R.: Variability in nocturnal nitrogen oxide processing and its role in regional air quality, Science, 311, 67-70, DOI 10.1126/science.1120120, 2006.*

Response: The method used to derive the $N_2O_5$ uptake coefficient in our manuscript did not require an assumption of $NO_3$ radical being in steady state, but assumed that the change of

$NO_3$ and $N_2O_5$ concentrations was mainly caused by $NO_3/N_2O_5$ chemistry. The value of $\frac{d([N_2O_5]+[NO_3])}{dt}$ was not required to be nearly zero as the method of Brown et al., 2006, but was calculated with the measured concentration of $N_2O_5$ and the calculated concentration of $NO_3$. We believe our method allows more data for use in analysis than the steady-state approach.

Indeed we compare our method with the steady-state approximation (Brown et al., 2006) for calculation of the $\gamma_{N2O5}$ using equation (1) below. The plots of $\tau_{N2O5}^{-1}K_{eq}[NO_2]$ correlated to $0.25c_{N2O5}SaK_{eq}[NO_2]$ for four selected air masses in short-time periods which were proper to use the steady state assumption are presented in Figure 1 here. The $\gamma_{N2O5\text{-steady-state}}$ varied from 0.008 to 0.012 and was comparable to the uptake coefficients derived with the method in the manuscript in the same periods (see Table 1 here).

(1) $\tau_{N2O5}^{-1}K_{eq}[NO_2] \approx k_g + \frac{1}{4}C_{N2O5}\,Sa\,\gamma_{N2O5}K_{eq}[NO_2]$

[Figure]

Figure 1. Plots of $\tau_{N2O5}^{-1}K_{eq}[NO_2]$ versus $0.25c_{N2O5}SaK_{eq}[NO_2]$ for selected air masses.

Table 1. Comparison of $\gamma_{N_2O_5}$ derived with steady state method and with the method in the manuscript in the same periods.

| Date | $\gamma_{\text{N2O5-steady-state}}$ | $\gamma_{\text{N2O5-in the manuscript}}$ |
|---|---|---|
| Jan 4 20:26-21:56 | 0.009 | 0.011 |
| Jan 5 17:48-18:39 | 0.008 | 0.007 |
| Jan 6 19:55-21:00 | 0.008 | 0.009 |
| Jan 9 23:15-00:20 | 0.012 | 0.014 |

Without the need for steady-state assumption, we can make use of more observation data to derive the updated parameters.

*The uncertainty of the measured $N_2O_5$, NMHC and Sa and the overall uncertainty propagated to $N_2O_5$ uptake coefficient and $ClNO_2$ yield should be carefully evaluated and discussed. As the hygroscopic growth factor is hard to quantify for RH over 90%, the derived $N_2O_5$ uptake coefficient for those conditions may subject with larger uncertainties compared with other RH cases. This is an interesting point to be discussed.*

Response: The uncertainty of the measured $N_2O_5$ and NMHC was ±25% and ±20%, respectively, which will influence the item of $\frac{k_{NO_2+O_3}[NO_2][O_3]}{[N_2O_5]}$ and $\frac{\sum k_i[VOC_i]}{[NO_2]\times K_{eq}}$ in equation (2) here. According to the calculation in our manuscript, $k'_{N2O5}$ was two orders of magnitude higher than that of $\frac{\sum k_i[VOC_i]}{[NO_2]\times K_{eq}}$, suggesting the value of $\frac{k_{NO_2+O_3}[NO_2][O_3]}{[N_2O_5]}$ was far more than $\frac{\sum k_i[VOC_i]}{[NO_2]\times K_{eq}}$. Hence, the uncertainty of $N_2O_5$ uptake coefficient was mainly caused by the uncertainty of $N_2O_5$, $NO_2$ (±20%), $O_3$ (±5%) and Sa. The hygroscopic growth factor is hard to quantify for RH over 90%, thus the calculated Sa would present large uncertainty when RH reached over 90%. The average RH ranged from 59-85% during the selected periods in Table 2 in the revised manuscript. The uncertainty of Sa with RH below 90% was estimated to be around ±30% (Tham et al., 2016;Wang Z et al., 2017). The uncertainty of the calculated $N_2O_5$ uptake coefficient was then derived to be ±45%. The uncertainty of $ClNO_2$ yield was mainly caused by the uncertainty of $NO_2$ (±20%), $O_3$ (±5%) and $ClNO_2$ (±25%) and was derived to be ±30%.

(2) $k'_{N_2O_5} = \dfrac{k_{NO_2+O_3}[NO_2][O_3]}{[N_2O_5]} - \dfrac{d[N_2O_5]}{[N_2O_5]dt} - \dfrac{d(\frac{[N_2O_5]}{[NO_2]\times K_{eq}})}{[N_2O_5]dt} - \dfrac{\sum k_i[VOC_i]}{[NO_2]\times K_{eq}} = \dfrac{1}{4}C_{N_2O_5}S_a\gamma_{N_2O_5}$

Tham, Y. J., Wang, Z., Li, Q., Yun, H., Wang, W., Wang, X., Xue, L., Lu, K., Ma, N., Bohn, B., Li, X., Kecorius, S., Größ, J., Shao, M., Wiedensohler, A., Zhang, Y., and Wang, T.: Significant concentrations of nitryl chloride sustained in the morning: investigations of the causes and impacts on ozone production in a polluted region of northern China, Atmos. Chem. Phys., 16, 14959-14977, 10.5194/acp-16-14959-2016, 2016.

Wang, Z., Wang, W., Tham, Y. J., Li, Q., Wang, H., Wen, L., Wang, X., and Wang, T.: Fast heterogeneous $N_2O_5$ uptake and $ClNO_2$ production in power plant and industrial plumes observed in the nocturnal residual layer over the North China Plain, Atmos. Chem. Phys., 17, 12361-12378, 10.5194/acp-17-12361-2017, 2017.

In the manuscript, the following sentences were added:

Line 338-341: The uncertainty of the above $\gamma_{N2O5}$ was estimated to be ±45% due to the measurement uncertainty of $N_2O_5$ (±25%), $NO_2$ (±20%), $O_3$ (±5%) and $S_a$ (±30%). The uncertainty of $\phi_{ClNO2}$ was mainly caused by the uncertainty of $NO_2$ (±20%), $O_3$ (±5%) and $ClNO_2$ (±25%) and was estimated to be ±30%.

*The relationship between $N_2O_5$ uptake coefficient, $ClNO_2$ yield and the chemical properties of particles or the meteorological data (such as RH) should be investigated, especially in the part of text around Line 572 (table 1), the reason of the high gamma value in the Jan.3 17:40-20:50 should be addressed as which was much higher than other derived value.*

Response: We examined the correlation between $N_2O_5$ uptake coefficient, $ClNO_2$ yield and the concentrations of aerosol compositions or RH, and the results did not show any significant dependence of uptake coefficient/yield on these parameters. The below Table was added in the SI as Table S2.

Table 2. Average values (μg m$^{-3}$) of PM$_{2.5}$ loadings and the composition of PM$_{2.5}$ during the time periods corresponding to Table 2 in the revised manuscript.

| Date | Cl$^-$ | NO$_3^-$ | SO$_4^{2-}$ | NH$_4^+$ | OM | EC | PM$_{2.5}$ |
|---|---|---|---|---|---|---|---|
| Jan.3 17:40-19:00 | 0.9 | 19.7 | 8.8 | 6.5 | 37.4 | 8.0 | 86.4 |
| Jan 4 17:00-22:00 | 1.5 | 44.3 | 8.7 | 12.0 | 44.6 | 13.2 | 150.7 |
| Jan 5 17:00-22:00 | 1.6 | 68.9 | 15.5 | 15.3 | 56.6 | 14.2 | 216.6 |
| Jan 6 17:00-22:40 | 2.7 | 40.0 | 15.7 | 13.8 | 54.6 | 10.5 | 174.3 |
| Jan 9 19:00-00:20 | 0.8 | 29.9 | 7.2 | 8.9 | 36.7 | 11.6 | 117.3 |

Regarding the Jan 3rd case, the concentration of N$_2$O$_5$ and the Sa were both the lowest in the five cases, and the P$_{NO3}$ was the highest among all cases, leading to high N$_2$O$_5$ uptake coefficient. Taking a closer look at the data of that night, it can be divided into two periods with relatively high N$_2$O$_5$ of 200 pptv on average from 17:40-19:00 and low N$_2$O$_5$ of only 15 pptv on average from 19:10-20:50 (see Table 2 below and Figure 2 data in the red box). The second period was influenced more by fresh emission during the transportation of the air mass as indicated by the more variable NO$_2$ and O$_3$, making the calculation of $\gamma_{N2O5}$ and $\phi_{ClNO2}$ more difficult. In the revised manuscript, we decided to drop the second period, and the $\gamma_{N2O5}$ was 0.066 (17:40-19:00) in the Jan 3 case.

Table 3. Details for the two parts of the selected period from 17:40-20:50 on the night of Jan 3, 2017.

| Date | N$_2$O$_5$ pptv | Max-ClNO$_2$ pptv | NO$_2$ ppbv | O$_3$ ppbv | RH % | Sa μm$^2$ cm$^{-3}$ | P$_{NO3}$ ppbv h$^{-1}$ | k$_{NO3}$ 10$^{-3}$ s$^{-1}$ | L$_{N2O5}$ ppbv h$^{-1}$ | k$_{NO3}$/(Keq[NO$_2$]) 10$^{-5}$ s$^{-1}$ | k$_{N2O5}$ 10$^{-3}$ s$^{-1}$ | $\gamma_{N2O5}$ | $\phi_{ClNO2}$ |
|---|---|---|---|---|---|---|---|---|---|---|---|---|---|
| Jan.3 17:40-19:00 | 200 | 1029 | 20.0 | 77.8 | 59 | 2170 | 4.3 | 0.516 | 4.3 | 3.03 | 8.8 | 0.066 | 0.18 |
| Jan 3 19:10-20:50 | 15 | 3145 | 24.7 | 59.2 | 78 | 4970 | 3.5 | 0.840 | 3.5 | 3.68 | 41.2 | 0.162 | 0.36 |

[Figure]

Figure 2. Variation of $N_2O_5$, $ClNO_2$, $NO_3^-$, trace gases and meteorological conditions during the nighttime of Jan 3 to 4, 2017.

The related texts in the original manuscript were also revised carefully. The following sentences were added.

Line 327: The data show that the uptake coefficient ranged from 0.009 to 0.066.

Line 331-335: It is interesting to see much higher $\gamma_{N2O5}$ (0.066) on Jan 3 than those in other four nights (0.009-0.015), resulting from higher $P_{NO3}$ but much lower Sa and relatively low $N_2O_5$ concentrations on Jan 3. We examined known factors affecting the loss of $NO_3$ and $N_2O_5$ such as the concentrations of NO, NMHCs and aerosol compositions, but found no obvious difference between Jan 3 and other nights.

Line 341-343: The correlation between $\gamma_{N2O5}$, $\phi_{ClNO2}$ and the concentrations of aerosol compositions (see Table S2) or RH was investigated, and the results (not shown here) did not indicate any significant dependence of $\gamma_{N2O5}$ or $\phi_{ClNO2}$ on these parameters.

*Minor comments:*

*The description of the experimental setup of the key relevant parameters needs to be*

*strengthened, e.g. the limit of detection, the measurement uncertainties and measurement principle should be described.*

Response: Table 1 with the limit of detection, the measurement uncerinties and measurement principle was added in the manuscript.

Table 1. Technique, limit of detection, and uncertainty of measuring instruments for trace gases and aerosols.

| Species | Measurement techniques | Uncertainty | Detection limits |
|---------|------------------------|-------------|------------------|
| $ClNO_2$, $N_2O_5$ | CIMS | ±25% | 6 pptv |
| HONO | LOPAP | ±20% | 7 pptv |
| $O_3$ | UV photometry | ±5% | 0.5 ppbv |
| NO | Chemiluminescence | ±20% | 0.06 ppbv |
| $NO_2$ | Photolytical converter & Chemiluminescence | ±20% | 0.3 ppbv |
| $NO_y$ | MoO catalytic converter & Chemiluminescence | ±5% | <0.1 ppbv |
| $SO_2$ | Pulsed-UV fluorescence | ±5% | 0.1 ppbv |
| CO | IR photometry | ±5% | 4 ppbv |
| NMHCs | GC-FID/MS | ±15-20% | 20-300 pptv |
| OVOCs | DNPH-HPLC | ±1-15% | 20-450 pptv |
| $PM_{2.5}$ | MAAP | ±10% | <0.1 μg m$^{-3}$ |
| Aerosol Ions | GAC-IC | ±10% | 0.01-0.16 μg m$^{-3}$ |
| OC/EC | RT-4 SUNSET | ± 4-6% | 0.2 μg cm$^{-2}$ |

*Line 161. The reference of Yue et al., 2015 may not appropriate and suggests replacing by Dong et al., 2012 Dong, H. B., Zeng, L. M., Hu, M., Wu, Y. S., Zhang, Y. H., Slanina, J., Zheng, M., Wang, Z. F., and Jansen, R.: Technical Note: The application of an improved gas and aerosol collector for ambient air pollutants in China, Atmos Chem Phys, 12, 10519-10533, 10.5194/acp-12-10519-2012, 2012.*

Response: The suggested reference was adopted.

*Line 586 (figure 2) please check the data of wind speed, as the WS keep below 3 m s$^{-1}$ during the whole half month. And the plot style of $NO_y$ made the concentration of $NO_2$ hard to follow. The left y-axis of fourth panel should change to $PM_{2.5}$ or other more appropriate name.*

Response: We did not find problem with the wind speed data, and the wind speed data shown in the figure was 10 min average. The very low wind speeds in the observation period were consistent with the regional meteorological conditions presented in the pressure contour in the weather chart. We also investigated the regional wind speed in PRD during this period from some websites and found that low wind speed was a regional phenomenon. Our personnel on site in fact felt little wind flow during the period. The figure below shows the wind speed (5 min average) from Dec 23, 2016 to Jan 20, 2017 at Heshan site, and the wind speed reached 7 m s$^{-1}$ before Jan 2017. Therefore, the low wind speeds were real and conducive to the occurrence of the severe haze.

[Figure]

Figure 3. Wind speed from Dec 23, 2016 to Jan 20, 2017 at Heshan site.

The plot style of NO$_y$, NO$_2$ and NO was changed in the mentioned Figure. The left y-axis of fourth panel was changed to PM$_{2.5}$.

*The legend of the early night and later night in figure 6 and 7 should be explained. By the way, how about the NO$_3^-$ formation potential intercomparison in the day and night of Jan 9 to 10.*

Response: The periods in the early nighttime in Fig.6 and Fig.7 correspond to the periods in Table 2 in the revised manuscript. And the periods in the later nighttime correspond to the periods in Table 3 in the revised manuscript. So the captions of Fig.6 and Fig.7 were changed to make them better understood. The comparison of the NO$_3^-$ formation potential in the day and night of Jan 9 and 10 was not conducted due to the lack of data of NMHC after Jan 8 which made the model simulation of OH infeasible on the day of Jan 9. In the caption of Fig.7, the explanation was added as follows.

Line 680-684: Figure 6. Comparison between the measured $NO_3^-$ increase and the $NO_3^-$ formation potential in the early nighttime (periods in Table 2: Jan 3 17:40-19:00, Jan 4 17:00-22:00, Jan 5 17:00-22:00, Jan 6 17:00-22:40, Jan 9 19:00-00:20) and in the later nighttime (periods in Table 3: Jan 3-4 21:00-05:00, Jan 5 01:30-06:50, Jan 5-6 23:40-01:10, Jan 6-7 23:00-06:00, Jan 10 01:50-03:30).

Line 686-694: Figure 7. Comparison between the daytime (7:00 to 17:00 LT, assuming all gas phase $HNO_3$ partitioned into particle phase) and nighttime (17:00 to 7:00 LT of the next day) $NO_3^-$ formation potential. The early nighttime in each day represents the periods in Table 2, including Jan 3 17:40-19:00, Jan 4 17:00-22:00, Jan 5 17:00-22:00, Jan 6 17:00-22:40, and Jan 9 19:00-00:20. The later nighttime in each day represents the periods in Table 3, including Jan 3-4 21:00-05:00, Jan 5 01:30-06:50, Jan 5-6 23:40-01:10, Jan 6-7 23:00-06:00, and Jan 10 01:50-03:30. The intercomparison of the $NO_3^-$ formation potential in the day and night of Jan 9 and 10 was not conducted due to the lack of data of NMHC after Jan 8 which made the model simulation of OH infeasible on the day of Jan 9.

---

## Author Comment (AC2) · 12 Nov 2018

**The reviewers' comments are italicized followed by our responses and changes in manuscript shown in blue and red, respectively. And the corrections are also marked as red color in the revised manuscript.**

*Yun et al. present a suite of measurements related $N_2O_5$ formation and subsequent uptake to aerosol that take place in a semi-rural area of China. They show through interpretations of their measurements and some master chemical modeling that nocturnal $NO_x$ chemistry can likely account nearly 50% of aerosol nitrate mass loadings during these heavy pollution events.*

*This paper is written and presented well for the most part. The measurement methods portion is lacking even considering that an associated reference may describe additional details. Assuming my comments are appropriately addressed and some changes are made that would help to clarify the methods and the paper in general, I would recommend publication.*

Response: We appreciate the reviewer for the positive comments and helpful suggestions. The measurement method portion was rewritten and a table was added to present the detection limit and uncertainties for CIMS and other related instruments. More references were cited for details of the instruments.

Table 1. Technique, limit of detection, and uncertainty of measuring instruments for trace gases and aerosols.

| Species | Measurement techniques | Uncertainty | Detection limits |
|---|---|---|---|
| $ClNO_2$, $N_2O_5$ | CIMS | ±25% | 6 pptv |
| HONO | LOPAP | ±20% | 7 pptv |
| $O_3$ | UV photometry | ±5% | 0.5 ppbv |
| NO | Chemiluminescence | ±20% | 0.06 ppbv |
| $NO_2$ | Photolytical converter & Chemiluminescence | ±20% | 0.3 ppbv |
| $NO_y$ | MoO catalytic converter & Chemiluminescence | ±5% | <0.1 ppbv |
| $SO_2$ | Pulsed-UV fluorescence | ±5% | 0.1 ppbv |
| CO | IR photometry | ±5% | 4 ppbv |
| NMHCs | GC-FID/MS | ±15-20% | 20-300 pptv |
| OVOCs | DNPH-HPLC | ±1–15% | 20-450 pptv |
| $PM_{2.5}$ | MAAP | ±10% | <0.1 μg m$^{-3}$ |
| Aerosol Ions | GAC-IC | ±10% | 0.01-0.16 μg m$^{-3}$ |
| OC/EC | RT-4 SUNSET | ± 4-6% | 0.2 μg cm$^{-2}$ |

*Comments:*

*Line 97: recommend changing "highest ever reported value" to something that will age better like "largest reported value to date".*

Response: Adopted.

*Line 100: recommend changing "aerosol formation" to "aerosol nitrate formation"*

Response: Adopted.

*Line 128: the SI would be a great place to see the results of these instrument backgrounds and the extent to which they worked.*

Response: We added the relevant information on CIMS.

Line 127-129: Activated carbon packed in a filter was used to determine the instrument background which was 10.2 ± 2.2 and 8.9 ± 2.0 Hz on average for $N_2O_5$ and $ClNO_2$, respectively.

*Line 129: "standard gas of $N_2O_5$" sounds like you can purchase a standard cylinder of $N_2O_5$ (which you can't). Even if Wang et al. 2016 outlines these calibrations in more detail, a brief explanation is needed at the minimum. The description of these calibrations needs to be expanded and include $ClNO_2$ calibrations as well.*

Response: The part of "2.2 Chemical ionization mass spectrometer" in the manuscript was rewritten.

Line 129-134: In-situ offline calibration was carried out every day for $N_2O_5$ and every two days for $ClNO_2$ by mixing the respective synthetic standard into humidified zero air (with RH controlled at 60% in the present study). The $N_2O_5$ standard was generated by reacting excess $NO_2$ with $O_3$ and determined from the decrease of $NO_2$, and the $ClNO_2$ was synthesized by the uptake of a known concentration of $N_2O_5$ on a NaCl slurry (see Wang T et al., 2016 and Tham et al., 2016 for details).

*Line 132-133: How were detection limits calculated? What signal-to-noise was used, etc.? I think the authors only mean the uncertainty is +/- 25% not the precision.*

Response: The detection limit was 6 pptv for both $N_2O_5$ and $ClNO_2$. It is defined as the signal twice of noise for 1 min averaged data. The noise was the standard error of the 1-min background measurement. The uncertainty of the measurement was estimated to be ± 25 % for both $N_2O_5$ and $ClNO_2$ (Wang et al., 2016).

Wang, T., Tham, Y. J., Xue, L., Li, Q., Zha, Q., Wang, Z., Poon, S. C., Dubé, W. P., Blake, D. R., and Louie, P. K.: Observations of nitryl chloride and modeling its source and effect on ozone in the planetary boundary layer of southern China, J. Geophys. Res. Atmos., 121, 2476–2489, doi: 10.1002/2015JD024556, 2016.

Line 137-138: The detection limits of $N_2O_5$ and $ClNO_2$ were both 6 pptv (2 σ, 1 min-averaged data).

Line 150-151: The uncertainty of the measurement was estimated to be ± 25 % for both $N_2O_5$ and $ClNO_2$ (Wang T et al., 2016).

*Line 169: Here and throughout the paper it's probably best to change "aerosol surface*

*density" to "aerosol surface area density" for clarity's sake.*

Response: Adopted.

*Line 182: consider changing "calculate" to "estimate"*

Response: Adopted.

*Line 214: change "matters" to "matter"*

Response: Adopted.

*Line 265/284: k' is often used for a pseudo first order rate constant. Consider using that to help differentiate from other rate constants.*

Response: Adopted.

*Line 307/309: make pNO$_3^-$ and p(NO$_3^-$) consistent. Use one or the other. p(NO$_3^-$) is used in the rest of the paper.*

Response: p(NO$_3^-$) was used in all places of the paper.

*Line 313/320 and Figure 6: Do the authors have a firm definition of what is considered "early nighttime" vs "late nighttime"? What times correspond to each period? Are these the same as provided in Table 1?*

Response: The periods in the early nighttime in Fig.6 and Fig.7 correspond to the periods in Table 2 in the revised manuscript. And the periods in the later nighttime correspond to the periods in Table 3 in the revised manuscript. The captions of Fig.6 and Fig.7 have been changed to make them better understood.

Line 680-684: Figure 6. Comparison between the measured NO$_3^-$ increase and the NO$_3^-$ formation potential in the early nighttime (periods in Table 2: Jan 3 17:40-19:00, Jan 4 17:00-22:00, Jan 5 17:00-22:00, Jan 6 17:00-22:40, Jan 9 19:00-00:20) and in the later nighttime (periods in Table 3: Jan 3-4 21:00-05:00, Jan 5 01:30-06:50, Jan 5-6 23:40-01:10, Jan 6-7 23:00-06:00, Jan 10 01:50-03:30).

Line 686-694: Figure 7. Comparison between the daytime (7:00 to 17:00 LT, assuming all gas

phase $HNO_3$ partitioned into particle phase) and nighttime (17:00 to 7:00 LT of the next day) $NO_3^-$ formation potential. The early nighttime in each day represents the periods in Table 2, including Jan 3 17:40-19:00, Jan 4 17:00-22:00, Jan 5 17:00-22:00, Jan 6 17:00-22:40, and Jan 9 19:00-00:20. The later nighttime in each day represents the periods in Table 3, including Jan 3-4 21:00-05:00, Jan 5 01:30-06:50, Jan 5-6 23:40-01:10, Jan 6-7 23:00-06:00, and Jan 10 01:50-03:30.

*Line 343-345: please change "$cm^{-3}$" units to commonly used "$molec\ cm^{-3}$".*

Response: Adopted.

*Line 359: certainly this approach is relevant to areas outside of China as well? Suggest removing "China".*

Response: Adopted.

*Table 1: Addition average aerosol nitrate loadings and $PM_{2.5}$ loadings for these periods would very useful. Consider adding all aerosol data (sulfate, ammonium, OM, etc.).*

Response: A table for the average $PM_{2.5}$ loadings and the average concentrations of the main compositions of $PM_{2.5}$ were added into SI. We investigated the correlation between $N_2O_5$ uptake coefficient, $ClNO_2$ yield and the concentrations of aerosol compositions, and the results did not show any significant dependence of uptake coefficient/yield on any parameters.

Table S2. Average values ($\mu g\ m^{-3}$) of $PM_{2.5}$ loadings and the composition of $PM_{2.5}$ during the time periods corresponding to Table 2 in the revised manuscript.

| Date | $Cl^-$ | $NO_3^-$ | $SO_4^{2-}$ | $NH_4^+$ | OM | EC | $PM_{2.5}$ |
|---|---|---|---|---|---|---|---|
| Jan 3 17:40-19:00 | 0.9 | 19.7 | 8.8 | 6.5 | 37.4 | 8.0 | 86.4 |
| Jan 4 17:00-22:00 | 1.5 | 44.3 | 8.7 | 12.0 | 44.6 | 13.2 | 150.7 |
| Jan 5 17:00-22:00 | 1.6 | 68.9 | 15.5 | 15.3 | 56.6 | 14.2 | 216.6 |
| Jan 6 17:00-22:40 | 2.7 | 40.0 | 15.7 | 13.8 | 54.6 | 10.5 | 174.3 |
| Jan 9 19:00-00:20 | 0.8 | 29.9 | 7.2 | 8.9 | 36.7 | 11.6 | 117.3 |

*Figure 5: Why not include the other aerosol data in this figure? $NO_3^-$ does not track with Sa,*

*so what is driving up Sa? The other data should explain this.*

Response: We examined the measured aerosol composition data. Similar to nitrate, sulfate and ammonium did not show large increase, while $PM_{2.5}$ levels increased, contributing to the increase in Sa_dry by about 30% (see figure 1 below). The 5-fold rise in Sa_wet was mainly due to the RH increase from ~90% to nearly 100% leading to a sharp increase in the growth factor. For the calculation of $N_2O_5$ uptake coefficient and $ClNO_2$ yield in the five select cases, the high Sa values under RH more than 90% were not included.

[Figure]

Figure 1. The variation of RH and $PM_{2.5}$ concentrations during the night of Jan 4-5 in the upper panel, the ratio of Sa_wet/Sa_dry in the middle panel, and Sa under dry conditions and the calculated Sa considering the variation of RH in the lower panel.

*Have the authors considered boundary layer effects in any of their analyses? With a shallow nighttime inversion layer and little mixing many of these species could be further concentrated. Are there any measurements taken during the study that would give boundary layer information (sondes, etc.)?*

Response: PBL was not measured at the site but should affect the variation of trace gas and aerosol concentration. We have added the following text in the revision.

Line 242-250: Apart from chemical reactions, the evolution of the Planetary Boundary Layer (PBL) also affects the concentrations of trace gas and aerosols. The height of PBL generally

decreases after sunset with the faster drop in temperature of land, which could lead to the accumulation of primary pollutants (and secondary pollutants) at surface if significant local sources are present. For example, on the night Jan 4-5 (see Fig 5), the CO and $NO_y$ levels increased between 18:00-19:00 with enhancement of $ClNO_2$ and nitrate, indicative of accumulation of primary emissions, but afterward the primary pollutants decreased for three hours while the latter two continued to increase due to the nighttime chemical process.

---

## Author Comment (AC3) · 12 Nov 2018

**The reviewers' comments are italicized followed by our responses and changes in manuscript shown in blue and red, respectively. And the corrections are also marked as red color in the revised manuscript.**

*The manuscript "Nitrate formation from heterogeneous uptake of dinitrogen pentoxide during a severe winter haze in southern China" by Yun and Co-authors uses observations of aerosol and gases and model results to study the contribution of heterogeneous chemistry via $N_2O_5$ to nitrate formation in $PM_{2.5}$ during severe winter haze episodes.*

*The measurements were carried out in the rural site of Heshan, located near the Perl River Delta in Southern China. In addition to a comprehensive suite of measurements, Yun and co-authors present the results of a chemical box model to estimate daytime $HNO_3$ mixing ratios. The box model was constrained by observation and its results were used to assess the importance of nighttime $N_2O_5$ over daytime $HNO_3$ as source of secondary aerosol nitrate.*

*The paper is well written, well structured and conveys results of interest for the scientific community. However, the method section (as pointed out by Refereees #1 and #2 as well) has to be improved and that some more discussion on 1)uncertainties, 2)sensitivity test of the model and 3)boundary layer dynamic needs to be added before publication.*

Response: The description of the measurement method has been rewritten and some related references were added. A table presenting the detection limit and uncertainties for CIMS and other related instruments was added. Sensitivity tests were conducted by reducing 10% of the input concentrations of NMHCs to check the variation of the rate of $OH+NO_2$ during the daytime. We have added discussion on the role of boundary layer dynamics.

*1) The description of the measurements should include the detection limits and the uncertainties, in particular for the species that were used to constrain the chemical box model.*

Response: Table 1 with detection limits and measurement uncertainties was added in the revised manuscript.

Table 1. Technique, limit of detection, and uncertainty of measuring instruments for trace gases and aerosols.

| Species | Measurement techniques | Uncertainty | Detection limits |
|---|---|---|---|
| $ClNO_2$, $N_2O_5$ | CIMS | ±25% | 6 pptv |
| HONO | LOPAP | ±20% | 7 pptv |
| $O_3$ | UV photometry | ±5% | 0.5 ppbv |
| NO | Chemiluminescence | ±20% | 0.06 ppbv |
| $NO_2$ | Photolytical converter & Chemiluminescence | ±20% | 0.3 ppbv |
| $NO_y$ | MoO catalytic converter & Chemiluminescence | ±5% | <0.1 ppbv |
| $SO_2$ | Pulsed-UV fluorescence | ±5% | 0.1 ppbv |
| CO | IR photometry | ±5% | 4 ppbv |
| NMHCs | GC-FID/MS | ±15-20% | 20-300 pptv |
| OVOCs | DNPH-HPLC | ±1–15% | 20-450 pptv |
| $PM_{2.5}$ | MAAP | ±10% | <0.1 $\mu g\ m^{-3}$ |
| Aerosol Ions | GAC-IC | ±10% | 0.01-0.16 $\mu g\ m^{-3}$ |
| OC/EC | RT-4 SUNSET | ± 4-6% | 0.2 $\mu g\ cm^{-2}$ |

*2) There should be a discussion in the main text or in the SI about the sensitivity of the box model to the uncertainties of the measurements (this, for example should be communicated with uncertainty bars in figure 7).*

Response: Sensitivity tests were carried out by reducing the input concentrations by 10% to check the deviation of the average daytime (7:00-17:00) rate of $OH+NO_2$ reaction. The method of Relative Increment Reactivity (RIR) was applied here as the index of the sensitivity (see the following equation). $R_1$ means the original rate of $OH+NO_2$ reaction, while $R_{0.9}$ means the rate of $OH+NO_2$ reaction after the input concentrations were reduced to 90%.

$$RIR = \frac{(R_1 - R_{0.9})/R_1}{10\%}$$

NMHCs were categorized into four groups, including C4HC, LRHC, AROM and OLF, which

represent alkanes with ≥4 carbons, hydrocarbons with low reactivity (including ethane, propane and benzene), reactive aromatics (including all aromatics except for benzene), and reactive olefins (including all alkenes), respectively (Xue et al., 2014). From the following figure, the simulated rate of OH+NO$_2$ reaction was most sensitive to HONO (RIR of 0.6-0.8), followed by NO$_x$ (RIR of 0.2-0.5) and OVOCs (RIR of 0-0.2).

[Figure]

Figure 1. OBM-calculated RIRs to check the sensitivity of the average daytime (7:00-17:00) rate of OH+NO$_2$ reaction to the uncertainties of the measured input data.

Xue, L., Wang, T., Louie, P. K. K., Luk, C. W. Y., Blake, D. R., and Xu, Z.: Increasing external effects negate local efforts to control ozone air pollution: A case study of Hong Kong and implications for other chinese cities, Environmental Science and Technology, 48, 10769-10775, 10.1021/ es503278g, 2014.

Line 211-214: Sensitivity tests were carried out by reducing the input concentrations by 10% to check the deviation of the average daytime (7:00-17:00) rate of OH+NO$_2$ reaction. The simulated rate of OH+NO$_2$ reaction was most sensitive to HONO, followed by NO$_x$ and OVOCs (see Text. S1 and Fig. S2).

*3) In paragraph 2.2 a discussion about interference for species with the same nominal mass as I(N$_2$O$_5$)$^-$ and I(ClNO$_2$)$^-$ should be added. How much contribution from other species would Yun and Co-authors expect? If it was not negligible how would change the results from the box model/comparison?*

Response: To the best of our knowledge, no interference was reported for I(N$_2$O$_5$)$^-$ at 235 m/z in current publications. Besides, we compared ambient measurements of N$_2$O$_5$ using the

quadrupole CIMS and NOAA-CRDS in 2016 (Wang et al., 2016), and $N_2O_5$ measured by CIMS and CRDS matched well with each other (slope=0.99, $R^2$=0.93). Recent ambient measurement of $ClNO_2$ in Beijing with a Tof-CIMS showed that $I(HNO_3)(H_2O)^-$ may cause ~10% interference of $ClNO_2$ at 208 m/z (Breton et al., 2018), but this kind of interference cannot be resolved by a quadrupole CIMS. For the quadrupole CIMS, we checked the correlation between the measured signal at 208 m/z ($I^{35}ClNO_2^-$) and at 210 m/z ($I^{37}ClNO_2^-$) during the present field campaign. The slope (0.317, $R^2 = 0.99$) was very close to the theoretical value of chlorine isotopic ratio of 0.32. Overall, we do not expect large (>10%) interference to $ClNO_2$, and no known interference is known to the $N_2O_5$ signal.

Breton, M. L., Hallquist, Å. M., Pathak, R. K., Simpson, D., Wang, Y., Johansson, J., Zheng, J., Yang, Y., Shang, D., and Wang, H.: Chlorine oxidation of VOCs at a semi-rural site in Beijing: significant chlorine liberation from $ClNO_2$ and subsequent gas-and particle-phase Cl–VOC production, Atmospheric Chemistry and Physics, 18, 13013-13030, 2018.

Wang, T., Tham, Y. J., Xue, L., Li, Q., Zha, Q., Wang, Z., Poon, S. C., Dubé, W. P., Blake, D. R., and Louie, P. K.: Observations of nitryl chloride and modeling its source and effect on ozone in the planetary boundary layer of southern China, J. Geophys. Res. Atmos., 121, 2476–2489, doi: 10.1002/2015JD024556, 2016.

*4) Relative humidity (RH) is known to affect measurements carried out with I-CIMS. Was the inlet used in this study humidified? Was the RH controlled/monitored during zero measurements? How could the zero affect the box model results (e.g., over/under estimation of $N_2O_5/ClNO_2$)? What are the biases that the 6 m sampling line could generate in their results?*

Response: Similar to our previous practice, the effect of RH on the sensitivity of $N_2O_5$ and $ClNO_2$ was measured by altering the RH in calibration during the present campaign (see below figure). The sensitivity of $N_2O_5$ and $ClNO_2$ in ambient measurement was corrected based on the RH monitored in real-time in the CIMS inlet.

[Figure]

Figure 2. The sensitivity of CIMS as a function of RH for (a) $N_2O_5$ at 235 m/z and (b) $ClNO_2$ at 208 m/z at Heshan site.

The inlet in this study was not humidified. Since the sampling period in this study was humid enough (RH>40%) to form the reagent $I(H_2O)^-$, a humidified inlet was not necessary.

The RH was not controlled but monitored during zero (and ambient) measurements.

The zero signals were subtracted from the total signals during data processing, thus they do not affect final data and thus modeling results.

The 6 m sampling tubing was replaced every day in late afternoon. The wall loss of $N_2O_5$ was measured by injecting synthetic $N_2O_5$ each time before and after replacing the sampling tubing. The measured wall loss of $N_2O_5$ was ~10% for the clean tubing and increased to ~40% after one day's sampling. Because our analysis mainly focused on data in the first few hours of evening, the loss was insignificant and thus was not corrected in our final data. However, this bias can be important at later period before tube replacement.

In the revised manuscript, we have added description on dependence of sensitivity on RH and how to correct it and also the above figure as Fig. S1. We also added the following sentences.

Line 134-137: The average sensitivity of $N_2O_5$ and $ClNO_2$ was $0.9\pm0.3$ and $0.7\pm0.2$ Hz pptv$^{-1}$, respectively. The dependence of the sensitivity on the relative humidity was measured during the field study (see Fig. S1) which was used to correct for the RH effect based on the measured ambient RH values.

Line 145-150: The loss of $N_2O_5$ on the tubing wall was checked on site by injecting $N_2O_5$ into

the ambient air before and after the tubing replacement, and the loss was around 10% in the "clean" tubing and increased to nearly 40% in the next afternoon. Because our analysis mainly focused on data in the first few hours of evening, the loss was insignificant and thus was not corrected in our final data. However, this bias can be important at later period before tube replacement.

*5) The boundary layer plays a significant role in the time evolution of the concentrations of nitrate in the particle. Yun and Co-authors make little to no mention of its role. For example, one might expect that particulate nitrate would increase also in the early morning hours due to the contribution of the residual layer during the mixing. This doesn't seem to happen in the observations presented in this work. May the Authors discuss why that would be the case?*

Response: This point was also raised by other referees. The description of boundary layer dynamics has been added in the revised version. The absence of nitrate increase in the early morning in our study is consistent with previous observations at the site (Yue et al., 2015). It may be explained by enhanced evaporation of $NH_4NO_3$ to $HNO_3$ and $NH_3$ due to increased temperature.

Yue, D., Zhong, L., Zhang, T., Shen, J., Zhou, Y., Zeng, L., Dong, H., and Ye, S.: Pollution properties of water-soluble secondary inorganic ions in atmospheric $PM_{2.5}$ in the Pearl River Delta region, Aerosol Air Qual. Res, 15, 1737-1747, 2015.

*6) An increase in particulate nitrate concentrations (as well as $PM_{2.5}$) could also be due to a dilution effect (same magnitude of aerosol sources but reduced volume in which the aerosols are mixed). I recommend adding a few sentences explaining how the mixing of the residual layer in the morning hours could affect the results presented here.*

Response: The following discussion was added in the manuscript.

Line 242-250: Apart from chemical reactions, the evolution of the Planetary Boundary Layer (PBL) also affects the concentrations of trace gas and aerosols. The height of PBL generally decreases after sunset with the faster drop in temperature of land, which could lead to the accumulation of primary pollutants (and secondary pollutants) at surface if significant local sources are present. For example, on the night Jan 4-5 (see Fig 5), the CO and $NO_y$ levels

increased between 18:00-19:00 with enhancement of $ClNO_2$ and nitrate, indicative of accumulation of primary emissions, but afterward the primary pollutants decreased for three hours while the latter two continued to increase due to the nighttime chemical process.